# Assessment of the Appearance and Toxin Production Potential of Invasive Nostocalean Cyanobacteria Using Quantitative Gene Analysis in Nakdong River, Korea

**DOI:** 10.3390/toxins14050294

**Published:** 2022-04-21

**Authors:** Yong-Jin Kim, Hae-Kyung Park, In-Soo Kim

**Affiliations:** Nakdong River Environment Research Center, National Institute of Environmental Research, Daegu 43008, Korea; yjkim99@korea.kr (Y.-J.K.); factork@korea.kr (I.-S.K.)

**Keywords:** invasive nostocalean cyanobacteria, genus-specific primers, cyanotoxin primers, ddPCR, anatoxin-a, *Cuspidothrix*, *Cylindrospermopsis*, *Sphaerospermopsis*

## Abstract

Invasive nostocalean cyanobacteria (INC) were first reported in tropical regions and are now globally spreading rapidly due to climate change, appearing in temperate regions. INC require continuous monitoring for water resource management because of their high toxin production potential. However, it is difficult to analyze INC under a microscope because of their morphological similarity to nostocalean cyanobacteria such as the genus *Aphanizomenon*. This study calculates the gene copy number per cell for each target gene through quantitative gene analysis on the basis of genus-specific primers of genera *Cylindrospermopsis*, *Sphaerospermopsis*, and *Cuspidothrix*, and the toxin primers of anatoxin-a, saxitoxin, and cylindrospermopsin. In addition, quantitative gene analysis was performed at eight sites in the Nakdong River to assess the appearance of INC and their toxin production potential. Genera *Cylindrospermopsis* and *Sphaerospermopsis* did not exceed 100 cells mL^−1^ at the maximum, with a low likelihood of related toxin occurrence. The genus *Cuspidothrix* showed the highest cell density (1759 cells mL^−1^) among the INC. Nakdong River has potential for the occurrence of anatoxin-a through biosynthesis by genus *Cuspidothrix* because the appearance of this genus coincided with that of the anatoxin-a synthesis gene (*anaF*) and the detection of the toxin by ELISA.

## 1. Introduction

Climate change, caused by global warming, has negative effects on aquatic ecosystems. In particular, harmful cyanobacterial algal blooms (cyanoHABs) that cause harmful substances (such as toxins and off-flavors) are increasing in extent and frequency due to climate change, negatively impacting water resources [1,2]. Representative cyanobacterial genera that cause cyanoHABs are *Microcystis*, *Dolichospermum*, and *Aphanizomenon* [1,3,4]. CyanoHABs and the damage associated with them have been globally reported in many lakes and rivers [5,6,7]. *Microcystis* spp. is the most common cause of cyanoHABs that produces microcystins that cause liver damage, and adversely affects human and livestock health in the short or long term, so microcystins are managed in many countries [4]. Furthermore, in addition to cyanobacteria that cause cyanoHABs, cyanobacteria of the order Nostocales, first reported in tropical regions, are now spreading to temperate regions such as North America and Europe [8,9,10]. These cyanobacteria are referred to as invasive or exotic cyanobacteria. Invasive nostocalean cyanobacteria (INC) include genera *Cylindrospermopsis*, *Sphaerospermopsis*, *Cuspidothrix*, *Raphidiopsis*, and *Chrysosporum* [11,12,13,14,15,16]. Species of these genera reportedly produce toxins such as saxitoxin, cylindrospermopsin, and anatoxin-a [10,17].

Recently, occurrences of INC have been reported in Northeast Asian countries, such as China and Japan [18,19,20,21]. In South Korea, *Cylindrospermopsis (Raphidiopsis) raciborskii* [22], and *Sphaerospermopsis aphanizomenoides* [23] were detected in the Han River system. In Nakdong River, the target site of this study, 24 strains of four species, *C. raciborskii*, *S. aphanizomenoides*, *S. reniformis*, and *Cuspidothrix issatschenkoi*, were isolated [24]. In particular, anatoxin-a synthesis gene *anaF* was detected in two strains of *C. issatschenkoi* isolated from the Nakdong River, and toxin production was verified by enzyme-linked immunosorbent assay (ELISA). These INC, which could produce toxins, are spreading globally and thus require continuous monitoring as part of water resource management. However, they are difficult to analyze using a microscope because they are morphologically similar to noninvasive nostocalean cyanobacteria such as *Aphanizomenon* and *Dolichospermum* [24,25]. In the case of *C. raciborskii*, a species-specific primer was developed [26], and molecular biological monitoring using real-time PCR analysis was conducted [27,28]. However, most studies on INC primarily comprise morphological microscopic analyses [16,29,30] or only focus on toxin-related genes [28,31,32]. Recently, genus-specific primers that can detect five genera of INC (*Cuspidothrix*, *Cylindrospermopsis*, *Sphaerospermopsis*, *Raphidiopsis*, and *Chrysosperum*) were developed [33]. These INC genus-specific primers can effectively detect these genera. Moreover, the shortcoming of the species-specific primer for *C. raciborskii* developed by Wilson et al. [26], which could not identify the *Raphidiopsis* genus, was improved.

Monitoring INC should include analyses of their toxin generation potential and target algal appearance. The toxin production potential of INC can be effectively determined by performing quantitative gene analysis on the basis of existing toxin-detection [24,34,35,36,37] and genus-specific primers. Therefore, this study develops a quantitative gene analysis method on the basis of genus-specific and toxin-detection primers for INC. We also investigate INC and toxin production potential in a temperate region by analyzing the appearance trend of INC and cyanobacteria with toxin-related genes at eight sites in Nakdong River. We expect it to be used for water supply management through using the molecular and ecological monitoring system developed in this study, and that the INC and related toxin production potential according to climate change can be more precisely analyzed than with morphological analysis using a microscope.

## 2. Results

### 2.1. Genus-Specific Gene Copy Numbers per Cell of INC

The gene copy numbers of three species of INC were proportional to the cyanobacterial cell count, with a high correlation (*r*^2^ = 0.91 or higher) (Figure 1). Quantitative analysis of the *rpoB* gene for 27–205 cells of the trace analysis sample of *Cuspidothrix* indicated 32–237 copies. The result of quantitative gene analysis for 933–9950 cells mL^−1^ of the sample diluted according to cell density ranged between 2312 and 13,480 copies mL^−1^, and the copy number of *rpoB* per cell was 1.4 copies cell^−1^. The copy numbers of *rpoB* per cell for the 11 strains of *Cuspidothrix* isolated from Nakdong River [24] were in the range of 1.0–1.4 copies cell^−1^, which were roughly similar to those for the NIVA-711 strain isolated in Germany, with an average of 1.1 copies cell^−1^ (Table 1).

Quantitative analysis of the *rbcLX* gene of *Sphaerospermopsis* for 24–211 cells in the trace analysis sample indicated 40–254 copies. The quantitative gene analysis result for 1068–9990 cells mL^−1^ of the analysis sample diluted by cell density indicated 1613–23,287 copies mL^−1^, and the copy number per cell of *rbcLX* was found to be 1.6 copies cell^−1^ (Figure 1). The strains of *Sphaerospermopsis* were isolated from Nakdong River. The copy numbers per cell of *rbcLX* for the eight remaining strains were in the range of 1.6–4.3 copies cell^−1^. Each strain showed a different copy number per cell, with an average of 3.0 copies cell^−1^ (Table 1).

For *Cylindrospermopsis*, the result of quantitative *rpoC1* gene analysis for 29–204 cells of the trace analysis sample indicated 106–253 copies. The result of quantitative gene analysis for 463–7420 cells mL^−1^ of the analytical sample per density indicated 1130–16,593 copies mL^−1^, and the copy number of the *rpoC1* gene per cell was 2.3 copies cell^−1^ (Figure 1). In the case of the four strains of *Cylindrospermopsis* isolated from Nakdong River, *rpoC1* gene copy numbers per cell were in the range of 1.2–1.5 copies cell^−1^. Strains isolated from Nakdong River did not show significant differences, but showed a difference of approximately 1 copy cell^−1^ from the CS-1101 isolated in Australia. Lastly, the copy number per cell for molecular ecological monitoring was calculated on the basis of the average copy number for each genus (Table 1). The average copy number for *Cylindrospermopsis* was based on the average copy number per cell for the four strains isolated from the Nakdong.

### 2.2. Toxin Gene Copy Numbers per Cell and Concentrations of Toxin-Producing Cyanobacteria

The gene copy numbers and concentrations of the three toxins (anatoxin-a, cylindrospermopsin, and saxitoxin) were measured. Results showed that, when the cell density of the NIVA-711 strain of *Cuspidothrix*, which produces anatoxin-a, was in the range of 933–9950 cells mL^−1^, the copy number of the *anaF* gene was 3080–14,580 copies mL^−1^, and the toxin concentration was 0.020–0.134 µg L^−1^. When the cell density of the CS-1101 strain of *Cylindrospermopsis*, which produces cylindrospermopsin, was in the range of 463–7420 cells mL^−1^, the copy number of the *cyrA* gene was 967–15,933 copies mL^−1^, the copy number of the *cyrJ* gene was 1664–21,400 copies mL^−1^, and the toxin concentration was 0.012–0.043 µg L^−1^. When the cell density of the NIVA-851 strain of *Aphanizomenon*, which produces saxitoxin, was in the range of 1826–14,620 cells mL^−1^, the copy number of the *sxtA* gene was 4280–23,866 copies mL^−1^, the copy number of the *sxtI* gene was 4347–22,800 copies mL^−1^, and the toxin concentration was 0.001–0.016 µg L^−1^. The three toxin genes showed proportional relationships with the cell counts and toxin concentrations (*r*^2^ = 0.96 or higher) (Figure 2). For anatoxin-a, the copy number of the *anaF* gene was 2.1 copies cell^−1^, and the toxin concentration was 0.012 pg cell^−1^. For cylindrospermopsin, the average copy number of the *cyrA* and *cyrJ* genes was 2.0 copies cell^−1^, and the toxin concentration was 0.013 pg cell^−1^. For saxitoxin, the average copy number of the *sxtA* and *sxtI* genes was 2.1 copies cell^−1^, and the toxin concentration was 0.001 pg cell^−1^ (Table 2).

### 2.3. Cell Density of INC in the Nakdong River

The genus *Cuspidothrix* was detected at a density of 0–1305 cells mL^−1^ during microscopic analysis. Quantitative gene analysis using a genus-specific primer (*rpoB*) showed a copy number of 0–1935 copies mL^−1^, which when converted into cell count, indicated 0–1759 cells mL^−1^ (Figure 3). *Cuspidothrix* showed the highest cell density among the three INC genera. Cell density was the highest in June at the upstream sites (SJ, ND, GM), and in August at the midstream and downstream sites (GG, DS, CH, HC). The highest cell density among all sites was detected at the ND site in the second week of June. During the same period, cell density was also high at the GM site, at 956 cells mL^−1^. At the ND site, density was 756 cells mL^−1^ even in the first week of June. In the second week of August, *Cuspidothrix* was detected at a concentration of approximately 200 cells mL^−1^ at the GG, DS, HC, and CH sites. At other sites, concentrations were less than 100 cells mL^−1^. Microscopic analysis showed that *Cuspidothrix* first appeared at the SJ and ND sites in the fourth week of May. However, quantitative gene analysis detected it at all locations in the first week of March. Microscopic and quantitative gene analysis results showed that *Cuspidothrix* could be observed at most sites until November.

The genus *Cylindrospermopsis* appeared in the range of 0–60 cells mL^−1^ in morphological microscopic analysis. The result of quantitative gene analysis using a genus-specific primer (*rpoC1*) showed a copy number in the range of 0–90 copies mL^−1^, corresponding to 0–69 cells mL^−1^ when converted into cell count. By site, the highest cell density was detected at GG in the third week of July. *Cylindrospermopsis* was detected at a concentration of 22–57 cells mL^−1^ at the GG and GM sites in the same period, at the ND site in the first week of April and the fourth week of October, and at the CH site in the second week of April. All other sites showed cell densities of less than 20 cells mL^−1^ (Figure 3).

The genus *Sphaerospermopsis* was detected at a concentration of 0–18 cells mL^−1^ under microscopic morphological analysis. The result of quantitative gene analysis using a genus-specific primer (*rbcLX*) showed a copy number in the range of 0–130 copies mL^−1^, corresponding to 0–44 cells mL^−1^ when converted into cell count. By site, the highest cell density was detected at the CH site in the third week of June, and other sites showed cell densities less than 20 cells mL^−1^ (Figure 3). Microscopic analysis results showed that the cell density of *Sphaerospermopsis* was low (less than 20 cells mL^−1^) at all sites, and that it first appeared at the GG site in the second week of July. However, the results of quantitative gene analysis showed that it was first detected at a concentration of less than 5 cells mL^−1^ in the first week of March at the SJ, ND, and DS sites, and then at low densities at all locations until the second week of June. Furthermore, *Sphaerospermopsis* was detected until the third week of September according to microscopic analysis. However, in gene analysis, it was only detected at the CH site in the fourth week of October, whereas in November, it was not detected at any site.

### 2.4. Cyanobacteria with Toxin Genes in the Nakdong River

The anatoxin-a synthesis gene (*anaF*) copy number was in the range of 0–295 copies mL^−1^, and when converted into a cell count, they were 0–140 cells mL^−1^. By site, it was the highest at the ND site in the second week of June. Cyanobacteria with the *anaF* gene were detected in the range of 26–74 cells mL^−1^ at the SJ and GM sites during the same period. Furthermore, it was detected in the range of 25–77 cells mL^−1^ at the HC and CH sites in the second week of April, and at the GG, DS, HC, and CH sites in the second week of August. In contrast, it was lower than 10 cells mL^−1^ at other sites. The copy numbers of saxitoxin synthesis genes (*sxtA*, *sxtI*) were in the range of 0–101 copies mL^−1^, and when converted into cell count, they were 0–48 cells mL^−1^. By site, cell density was the highest at the ND site in the second week of July. Densities greater than 20 cells mL^−1^ were detected at the GM, HC, and CH sites from April to June. In contrast, densities were lower than 10 cells mL^−1^ at other sites. The copy numbers of the cylindrospermopsin synthesis genes (*cyrA*, *cyrJ*) were in the range of 0–46 copies mL^−1^, corresponding to 0–23 cells mL^−1^ when converted into cell count. By site, cell density was the highest at the GM site in the third week of July; except when 17 cells mL^−1^ was detected at the SJ site in the fourth week of June, the cell density was generally lower than 10 cells mL^−1^ at other sites (Figure 4).

### 2.5. ELISA-Based Toxin Analysis

The three types of toxins were analyzed using an ELISA kit at eight sites in the Nakdong River. Results showed that saxitoxin was detected in trace amounts in the range of 0.023–0.043 ng L^−^^1^ at the CG, GG, DS, and CH sites in April and May. Anatoxin-a was detected in the range of 0.154–0.284 ng L^−1^ at the ND and GM sites in the first and second weeks of June, but not at any other sites. Cylindrospermopsin was not detected at any site during the survey period (Table 3).

### 2.6. Appearance Characteristics of INC

All three genera of INC were detected at all eight sites in Nakdong River in March, and they remained until November (Figure 3). Regarding the appearance trend according to the water temperature range, *Cylindrospermopsis* was first detected at water temperatures below 10 °C, and showed the highest appearance frequency at water temperatures of 20–30 °C; cell density was the highest at a water temperature of approximately 20 °C. *Sphaerospermopsis* was first detected at water temperatures below 10 °C and showed the highest appearance frequency at water temperatures of 20–25 °C; the cell density was the highest at a water temperature of approximately 25 °C (Figure 5).

*Cuspidothrix* appeared in the full water temperature range (from below 10 to 30 °C). *Cuspidothrix* was also observed during phytoplankton monitoring analysis using a microscope at the eight weir sites in the Nakdong River in 2018–2019 [24]. Cell density was the highest at water temperatures of 20–25 °C and at the upstream sites of SJ, ND, and GM (Figure 6). At the ND site, where cell density was high, *Cuspidothrix* appeared at a concentration of 2524 cells mL^−1^ at water temperatures above 25 °C in June 2019. It began to appear at concentrations greater than 500 cells mL^−1^ after May, when the water temperature was maintained at 20 °C or higher.

## 3. Discussion

In this study, we developed a quantitative gene analysis method on the basis of genus-specific primers and toxin detection primers for INC, and they could be detected down to one trichome (approximately 25 cells). Additionally, INC could be detected down to the level of 1 cell mL^−1^ in field samples. Quantitative gene analyses in most prior studies targeted toxin genes of cyanobacteria that proliferate in large numbers, such as those in the genus *Microcystis* [38,39]. A few quantitative gene analysis studies were specifically conducted on *C. raciborskii*, for which a species-specific primer was developed [27,28,40]. However, these studies calculated the copy number per milliliter or liter for the target gene, and compared it with the analyzed cell count from field samples with a microscope [27,28] or calculated the cell count for a culture sample with a higher density than 1.2 × 10^7^ cells L^−1^ [40]. Therefore, it was difficult for them to accurately determine the cell density of target cyanobacteria that actually appear in rivers and lakes. Moreover, previous studies did not consider detecting target cyanobacteria in low-density samples.

Orr et al. [27] conducted a *C. raciborskii* monitoring study using real-time PCR. The copy number per cell of *rpoC1* was in the range of 1.3–19.0 copies cell^−1^, showing a large error. They analyzed samples from a lake rather than from cultured samples. Furthermore, morphological analyses using a microscope may underestimate the cell count of *Cylindrospermopsis* spp. when differentiated cells are not identified, such as the formation of terminal heterocytes and akinetes, which are classification keys, because of their morphological similarities with *Aphanizomenon* spp. and *Cuspidothrix* spp. [25]. This scenario seems to have caused differences in the copy number per cell. The difference in the copy number per cell between the *Cylindrospermopsis* strains isolated from Nakdong River and the CS-1101 strain isolated in Australia was not significant. However, there is a possibility of geographical heterogeneity [41]. Therefore, it was considered reasonable to calculate the cell count on the basis of the copy number per cell for strains isolated from the Nakdong River water system.

Quantitative gene analysis results could not be compared for *Cuspidothrix* and *Sphaerospermopsis* owing to the lack of prior studies. However, for *Cuspidothrix*, there were no significant differences in the copy numbers per cell for the genus-specific gene *rpoB* among strains. Hence, the analytical method used in this study may have helped in analyzing the cell density of *Cuspidothrix*. In the case of *Sphaerospermopsis*, genus-specific gene *rbcLX* showed differences in the copy numbers per cell for each strain. However, the average copy number per cell for the isolated strains was used because there were no primers that could specifically detect the target genus. Thus, additional development or improvements of the genus-specific primers is required.

Quantitative gene analysis results using toxin-detecting primers showed a proportional relationship between the gene copy numbers of the three toxins and the concentrations of each toxin, and cyanobacterial cell counts. However, the concentration of toxins is likely to change depending on environmental conditions, such as the growth stage of the target alga, the availability of nutrients, and water temperature [27,28]. Hence, accurate toxin concentration cannot be calculated using the quantitative toxin gene analysis results. However, this can be used as a proxy of the toxin production potential.

From the quantitative gene analysis for samples collected at eight sites in the Nakdong River, three genera of INC were detected at all study sites from March to November. Their appearance in the low-water-temperature period (March to April), which could not be verified by microscopic analysis owing to morphological similarities and the detection of only trace amounts, was confirmed using quantitative gene analysis. In the case of *C. raciborskii*, the minimal germination water temperature of the akinete was 15 °C [42]. However, in a lake in Poland, vegetative cells of *C. raciborskii* were detected at water temperatures below 7 °C [30]. In this study, *rpoC1* (*Cylindrospermopsis*) was detected at a water temperature of 8.3 °C in the second week of March. There is a possibility that the actual water temperature for the germination of the akinete could be lower than this. Genera *Cuspidothrix* and *Sphaerospermopsis* were detected at water temperatures below 7 °C at the time of the first survey (March 2). The vegetative cells of *S. aphanizomenoides* were observed at water temperatures below 10 °C in a previous study [24]. Furthermore, the lowest water temperature during the survey period was 6.9 °C, and the possibility of INC presence from December to February, which was not investigated in the present study, cannot be excluded.

In 2020, the highest temperature of Nakdong River rarely exceeded 30 °C because rainfall was 1.5–1.9 times greater than the average. Thus, the generation trend and appearance of INC at water temperatures higher than 30 °C could not be verified. However, among the three genera of INC, *Cuspidothrix* showed relatively higher cell density, and the difference between the microscopic analysis result and the converted cell count from the quantitative gene analysis was not substantial when more than 200 cells mL^−1^ appeared. Therefore, it was possible to compare results with past monitoring data on the basis of microscopic analysis. In Nakdong River, blooms are mainly generated by *Microcystis* spp., and large blooms of up to 800,000 cells mL^−1^ that are generated in summer and autumn [43], mainly occur in the midstream and downstream sections. The cell density of the genus *Cuspidothrix* was higher in upstream sites (SJ, ND, GM), where the cell density of *Microcystis* was smaller than that in the mid and downstream sites (Figure 6). During the period when the highest level of *Cuspidothrix* was detected in 2018–2019, cell density was the highest at a water temperature of 25 °C, similar to the result in 2020. This seemingly occurred because, although *Cuspidothrix* can grow at low water temperatures and actively proliferates at water temperatures above 20 °C, in summer, it is eliminated because of competition with *Microcystis* spp., a dominant cyanobacterium in the Nakdong River.

Quantitative analysis of the toxin genes showed that the cell count of the cyanobacteria with saxitoxin and cylindrospermopsin synthesis genes did not exceed 50 cells mL^−1^ at the maximum, and their appearance frequency was also low. In addition, the detection of the two toxin genes did not agree with the appearance trends of *Cylindrospermopsis* or *Sphaerospermopsis.* Thus, the toxin production potential of these two genera was considered to be low in the Nakdong River water system. Cylindrospermopsin was not detected in toxin measurements using ELISA, whereas saxitoxins were detected in trace amounts in the midstream and downstream sections in April and May. Although *Cuspidothrix* were detected during this period, there was no saxitoxin-producing strain among those isolated from the Nakdong River. Cyanobacteria with the *anaF* gene showed the highest cell count among the cyanobacteria with the three types of toxin genes; however, it was not found at concentrations exceeding 200 cells mL^−1^. The appearance of cyanobacteria with the *anaF* gene agreed with the appearance trend of *Cuspidothrix*, and with the time of detection of trace amounts of anatoxin-a by ELISA (Figure 3 and Figure 4; Table 3). A previous study found *anaF* gene expression and anatoxin-a production in *Cuspidothrix* strains (NRERC-654, 661) isolated from Nakdong River [24]; thus, *Cuspidothrix* was lastly verified as the alga that produced anatoxin-a in Nakdong River.

Although the appearance of INC and the amount of produced toxins were low, quantitative gene analysis using toxin- and genus-specific primers in Nakdong River revealed the potential production of anatoxin-a by *Cuspidothrix* at eight sites in Nakdong River. INC are currently increasing globally due to climate change, and many countries, including Korea, manage only microcystin, so it is necessary to manage toxins that INC can produce. Therefore, INC must be continuously monitored as part of the safe water quality management of Nakdong River, which is used as an important water source.

## 4. Conclusions

Quantitative gene analysis using toxin- and genus-specific primers enabled the precise analysis of the appearance times, cell densities, and toxin production potential of target algae. The toxin production potential of *Cuspidothrix*, rather than *Cylindrospermopsis*, was verified in Nakdong River, a temperate climate zone. The presence of *Cuspidothrix* must be continuously monitored for managing the water quality of Nakdong River, a crucial water source. Furthermore, the isolation of more INC strains and comprehensive analyses of gene copy numbers per cell for each strain are warranted to improve the accuracy of future quantitative gene analyses. If primers are developed and applied to cyanobacteria that produce various toxins, we expect that harmful cyanobacteria and related toxins can be monitored quickly and effectively using molecular biological techniques rather than microscopic analysis.

## 5. Materials and Methods

### 5.1. Selection of INC and Cultivation of Toxin-Producing Cyanobacteria

Three species of INC, *C. raciborskii*, *S. aphanizomenoides*, and *C. issatschenkoi*, were selected as the target species. In addition, *S. reniformis*, which had been found in the Nakdong River, was also included. However, *Raphidiopsis* and *Chrysosporum*, which are tropical and invasive cyanobacteria genera, were excluded from the target INC species in this study because their appearance in Nakdong River was not verified. For the cyanobacterial toxins, three toxins produced by INC, anatoxin-a, cylindrospermopsin, and saxitoxin, were selected: toxin-producing *C. issat**s**chenkoi* strain NIVA-711 (Oslo, Norway), which contains toxin synthesis gene *anaF*, was used for anatoxin-a; *C. raciborskii* strain CS-1101 (ANACC, CSIRO, Clayton South, Australia), which contains toxin synthesis genes *cyrA* and *cyrJ*, was used for cylindrospermopsin; and toxin-producing *Aphanizomenon gracile* strain NIVA-851 (Oslo, Norway) strain, which contains toxin synthesis genes *sxtA* and *sxtI*, was used for saxitoxin. The INC of 4 species and 24 strains [24] isolated from Nakdong River, and positive controls of 3 species and 3 strains, were cultured under controlled conditions at 20 °C, 40 µEm^−^^2^ s^−^^1^, and 14/10 h (Light/Dark) in MLA [44] medium. Strains were proliferated until the logarithmic growth phase before being used for quantitative gene analysis (Table 4).

### 5.2. Isolation of Genomic DNA

To extract genomic DNA, cultured algal and field samples were filtered with a 0.45 µm MicronSep nitrocellulose membrane disk (GVS Life Sciences, Findley, OH, USA) and stored at −80 °C until use. Genomic DNA was extracted using the QIAGEN DNeasy Plant Mini Kit (QIAGEN, Hilden, Germany). For the gene extraction process, a piece of filter paper stored at −80 °C was left at room temperature (15–25 °C) for 5 min. Then, a lysis buffer was added to the sample-containing filter paper, which was then sonicated (Vibra-Cell, Sonics & Materials, Newtown, CT, USA) following the protocol provided by the manufacturer. The concentration and purity of the extracted genomic DNA were measured using an Infinite M200 PRO Microplate Reader (Tecan Austria GmbH, Grödig, Austria). The extracted genomic DNA was stored at −20 °C.

### 5.3. Design of Toxin and Genus-Specific Primers for Quantitative Gene Analysis

In this study, quantitative gene analysis was performed using digital droplet PCR (ddPCR). This method offers the advantage of precisely detecting target genes in a complex mixture. It has been recently used to quantitatively analyze harmful marine algae [45] and microbes in water treatment plants [46], and to detect toxic cyanobacteria in plant leaves [47]. For quantitative gene analysis, the primer set was designed to include probes in addition to forward and reverse primers based on primers for detecting genes (*anaF*, *cyrA*, *cyrJ*, *sxtA*, and *sxtI*) related to the production of anatoxin-a, cylindrospermopsin, and saxitoxin [24,34,35,36,37] and genus-specific primers [33] of INC (*Cylindrospermopsis*, *Sphaerospermopsis*, and *Cuspidothrix*) (Table 5 and Table 6).

### 5.4. Quantitative Gene Analysis

The QX200 ddPCR system (Bio-Rad Laboratories Inc., Hercules, CA, USA), comprising a T100 Thermal Cycler measures the target gene copy number in single cells. When ddPCR was performed, the final sample volume was 20 μL. Each reaction mixture was composed of 10 μL of ddPCR^™^ Supermix for Probes (no dUTP) (Bio-Rad Laboratories Inc., Munich, Germany), 0.9 μL each of 10 pmol forward and reverse primers, 0.25 μL of 10 pmol probe, and 1 μL of sample DNA. The final volume was adjusted using sterile distilled water. Reaction mixtures were separately inserted into each well of a 96-well plate. Each reaction mixture was divided into 20,000 droplets of approximately 1 nm in the Bio-Rad QX200™ Droplet Generator (Bio-Rad Laboratories Inc., Hercules, CA, USA). Droplets were then transferred to a new 96-well plate. PCR was performed in a T100 Thermal Cycler (Bio-Rad Laboratories Inc.). After PCR, the plate was moved to a droplet reader (Bio-Rad Laboratories Inc.), and droplets were quantified by measuring their positive or negative signals. Signals were analyzed using QuantaSoft™ software version 1.7.4 (Bio-Rad Laboratories Inc.). Values were used as actual measurements only when at least 12,000 droplet signals were read [48]. The threshold values of the fluorescent signals were directly set according to the signal values of the negative and positive controls. All reagents related to ddPCR were purchased from Bio-Rad Laboratories, Inc.

### 5.5. Calculation of Genus-Specific Gene Copy Number per Cell for the Target Cyanobacteria

Quantitative gene analysis of INC was performed by converting the ddPCR result into a cell count. To this end, the copy number of genus-specific genes (*Cylindrospermopsis*: *rpoC1*, *Sphaerospermopsis*: *rbcLX*, *Cuspidothrix*: *rpoB*) per cell of the three genera of INC were analyzed. For the analysis of the genus-specific gene copy number per cell, representative strains were selected for each species (NIVA-711, NRERC-600, and CS-1101). Cultured cyanobacteria were diluted to a density of 1 × 10^3^, 3 × 10^3^, 5 × 10^3^, 7 × 10^3^, and 10 × 10^3^ cells mL^−1^, and filtered through a membrane disk to extract genomic DNA before performing ddPCR. Each diluted sample was preserved by adding Lugol’s iodine solution (final concentration 0.3%), and cell density was analyzed using a phase-contrast microscope (Imager M2; Carl Zeiss, Germany). Trace samples were analyzed by separating trichomes, the smallest unit of the target cyanobacteria, into 25, 50, 100, and 200 cells using a Pasteur pipette, and attaching them to a membrane filter. The gene copy number per cell (copies cell^−1^) was determined for each sample on the basis of cell density and quantitative gene analysis results. In addition to the representative strains, the copy number per cell for strains isolated from Nakdong River was calculated by performing density analysis.

### 5.6. Calculation of Toxin Gene Copy Number per Cell for Toxin-Producing Cyanobacteria

To calculate the toxin gene copy number per cell for toxin-producing cyanobacteria, strains that had been verified to produce toxins (CS-1101, NIVA-711, and NIVA-851) were cultured and diluted. Genomic DNA was extracted by filtering strains through a membrane disk, and ddPCR was performed for the toxin genes (anatoxin-a: *anaF*; cylindrospermopsin: *cyrA* and *cyrJ*; saxitoxin: *sxtA*, and *sxtI*). Each diluted sample was preserved by adding Lugol’s iodine solution, and cell density was analyzed using a phase-contrast microscope. Lastly, the toxin gene copy number per cell (copies cell^−1^) was calculated using the quantitative gene analysis results.

### 5.7. Analysis of INC at Eight Sites in Nakdong River

To analyze the cell density of the INC, 2 L of surface water was collected 500 m upstream from eight weir sites (Sangju, SJ; Nakdan, ND; Gumi, GM; Chilgok, CG; Gangjeong-Goryeong, GG; Dalseong, DS; Hapcheon-Changnyeong, HC; and Changnyeong-Haman, CH) from upstream to downstream of Nakdong River biweekly, from March to November (weekly from June to September) (Figure 7). The climate of the Nakdong River system is a temperate zone with four distinct seasons, which drop below 0 in winter, 10–15 °C in spring and autumn, and 25–30 °C in summer. In summer, typhoons and rainy seasons also bring much rainfall.

Collected samples were kept at 4 °C while being moved to the laboratory. For microscopic analysis, 1 L water samples were preserved in glass bottles with Lugol’s solution (0.3%). After leaving the sample for 48 h or more for natural precipitation, the supernatant was removed with a siphon and concentrated five to eight times. For quantitative gene analysis, 25 mL of a water sample was filtered using a 0.45 µm MicronSep™ nitrocellulose membrane disk to extract the genomic DNA, which was stored at −20 °C. For toxin analysis, water samples were used to fill 50 mL conical tubes, which were then stored at −20 °C. Before analysis, samples were thawed in a refrigerator at 4 °C, sonicated, and moved to separate tubes to individually analyze the three toxins. Anatoxin-a and saxitoxin were treated by adding the preservative solution included in the ELISA kit.

For morphological analysis of the INC using a microscope, 1 mL of the concentrated sample was analyzed at 100–400× magnification under a phase-contrast microscope using a Sedgewick Rafter counting chamber [25,49]. To identify the INC, only those cells that allowed for a clear distinction of the species by the characteristics of the terminal cells, the formation and position of the heterocytes, and the position of the akinete, which are important classification indicators for these species, were identified. Cell count per milliliter was determined by converting the concentration ratio.

For quantitative analysis using the genus-specific primers for INC, ddPCR was performed for genomic DNA using genus-specific primers (Table 5). Regarding the copy number of the target genes, the cell count of the target cyanobacteria per milliliter was calculated using the copy number per cell derived from quantitative gene analysis. For the quantitative analysis of cyanobacteria harboring toxin-related genes, ddPCR was performed for genomic DNA using the toxin primer set (Table 6). For the copy number of the target genes, the cell count per milliliter of cyanobacteria with toxin genes was calculated using the copy number per cell determined in this study. For the saxitoxin and cylindrospermopsin synthesis genes, cells were counted only when both pairs of genes (*sxtA* and *sxtI*, and *cyrA* and *cyrJ*) were detected. Regarding the anatoxin synthesis gene, it was considered that the toxin could only be produced if the *anaF* gene was present.

### 5.8. Toxin Analysis Using ELISA Kit

Toxin analysis of the cultured cyanobacterial strains and water samples collected near the eight weirs was performed using anatoxin-a (520060), saxitoxin (52255B), and cylindrospermopsin (522011) ELISA kits (Eurofins Abraxis, Inc., Warminster, PA, USA). For analysis, cultured cyanobacteria and water samples were placed in 50 mL tubes and sonicated on ice, followed by ELISA according to the manufacturer’s instructions. Absorbance was measured at 450 nm using an Infinite M200 PRO microplate reader (Tecan Austria GmbH, Grödig, Austria).

## Figures and Tables

**Figure 1 toxins-14-00294-f001:**
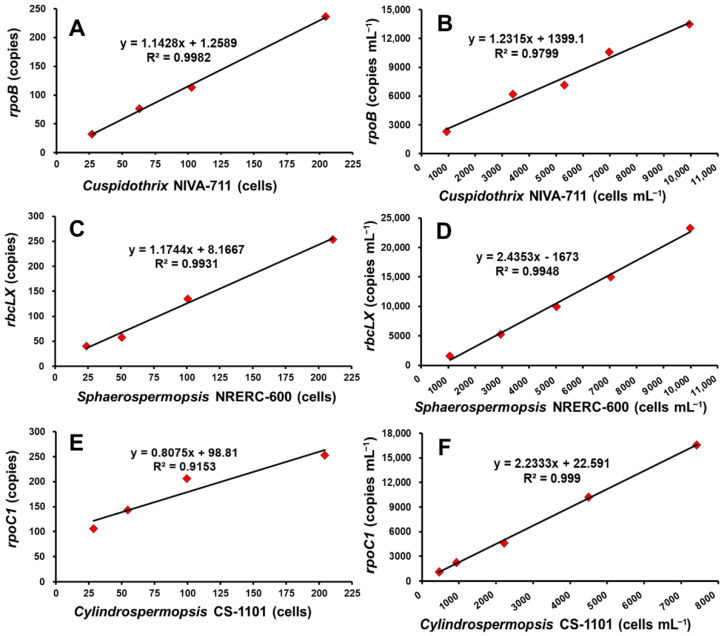
Relationship of cyanobacterial cell densities and gene copies using genus-specific primers. (**A**,**B**) *Cuspidothrix* NIVA-711; (**C**,**D**) *Sphaerospermopsis* NRERC-600; (**E**,**F**) *Cylindrospermopsis* CS-1101).

**Figure 2 toxins-14-00294-f002:**
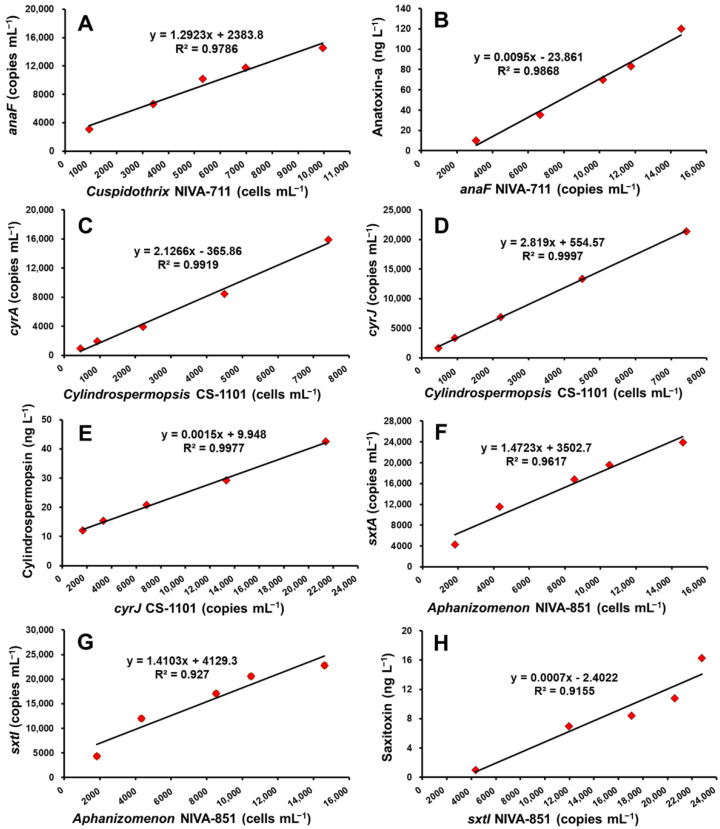
Relationship of cyanobacterial cell densities, gene copies using cyanotoxin primers, and cyanotoxin concentration. (**A**,**B**) *Cuspidothrix* NIVA-711, *anaF* gene copies and anatoxin-a; (**C**–**E**) *Cylindrospermopsis* CS-1101, *cyrA*, *cyrJ* gene copies and cylindrospermopsin; (**F**–**H**) *Aphanizomenon* NIVA-851, *sxtA*, *sxtI* gene copies and saxitoxin).

**Figure 3 toxins-14-00294-f003:**
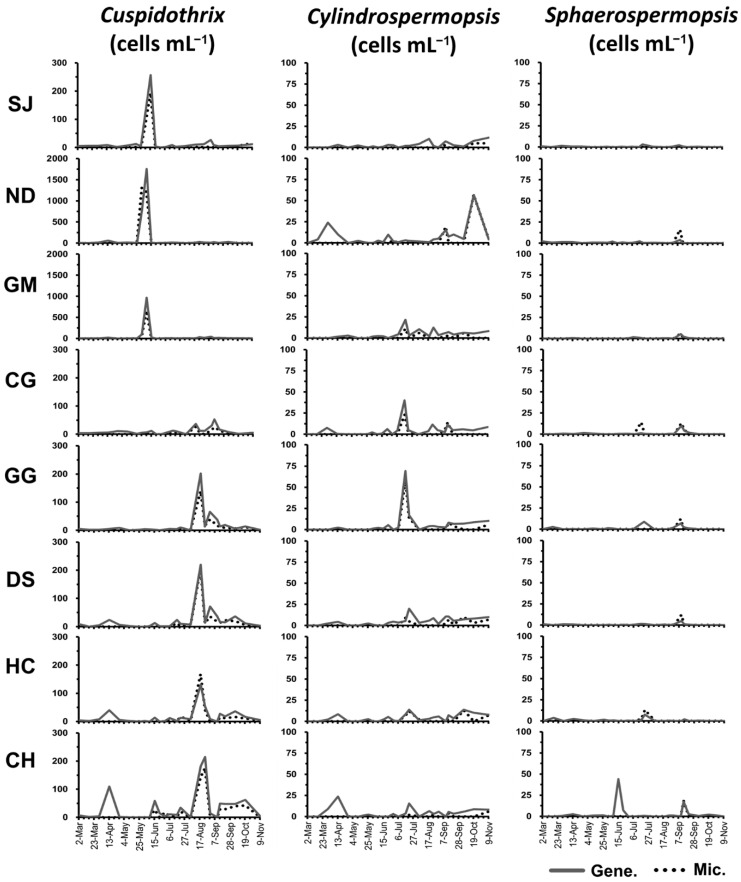
Status of appearance of invasive nostocalean cyanobacteria in the Nakdong River (2020) (Gene.: quantitative gene analysis, Mic.: microscopic analysis).

**Figure 4 toxins-14-00294-f004:**
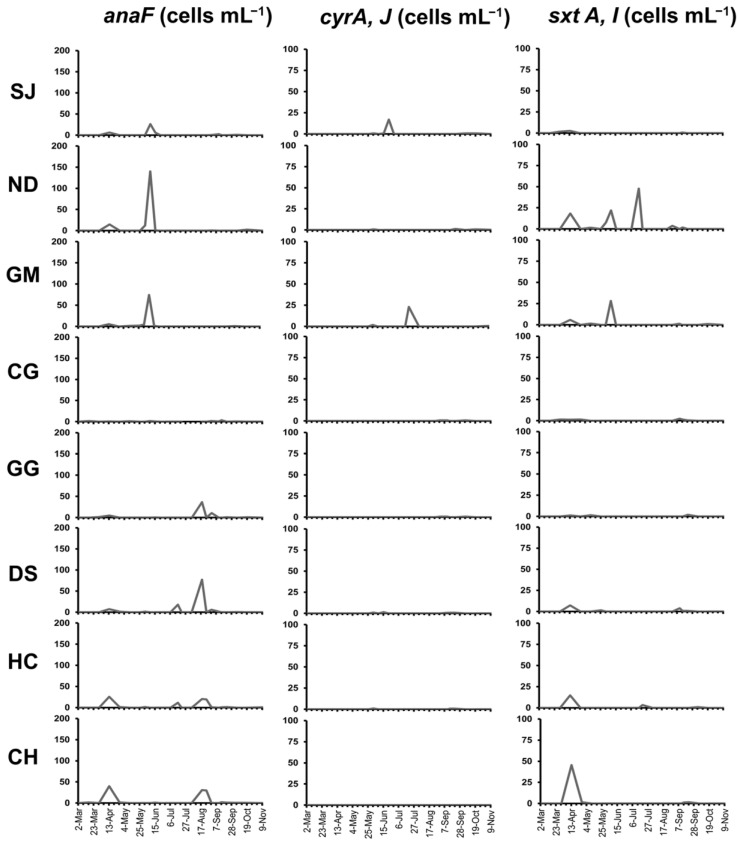
Result of quantitative gene analysis using toxin primer converted into cell number in Nakdong River (2020).

**Figure 5 toxins-14-00294-f005:**
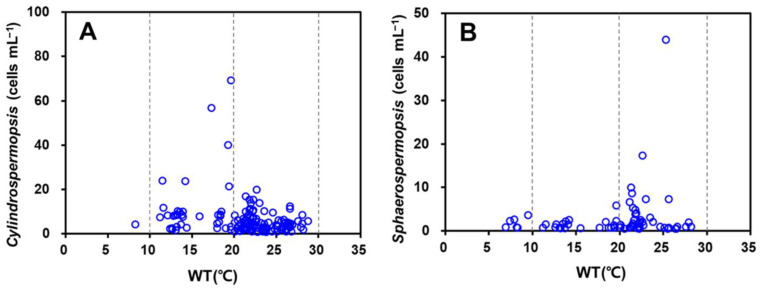
Appearance of genera (**A**) *Cylindrospermopsis* and (**B**) *Sphaerospermopsis* according to water temperature range (2020).

**Figure 6 toxins-14-00294-f006:**
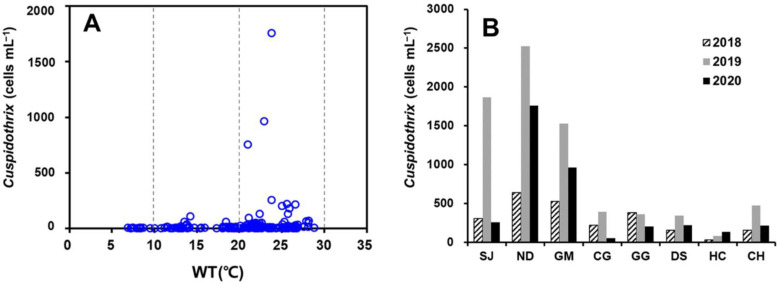
Appearance of genus *Cuspidothrix* according to (**A**) water temperature range (2020) and (**B**) maximal cell density (2018–2020) in Nakdong River.

**Figure 7 toxins-14-00294-f007:**
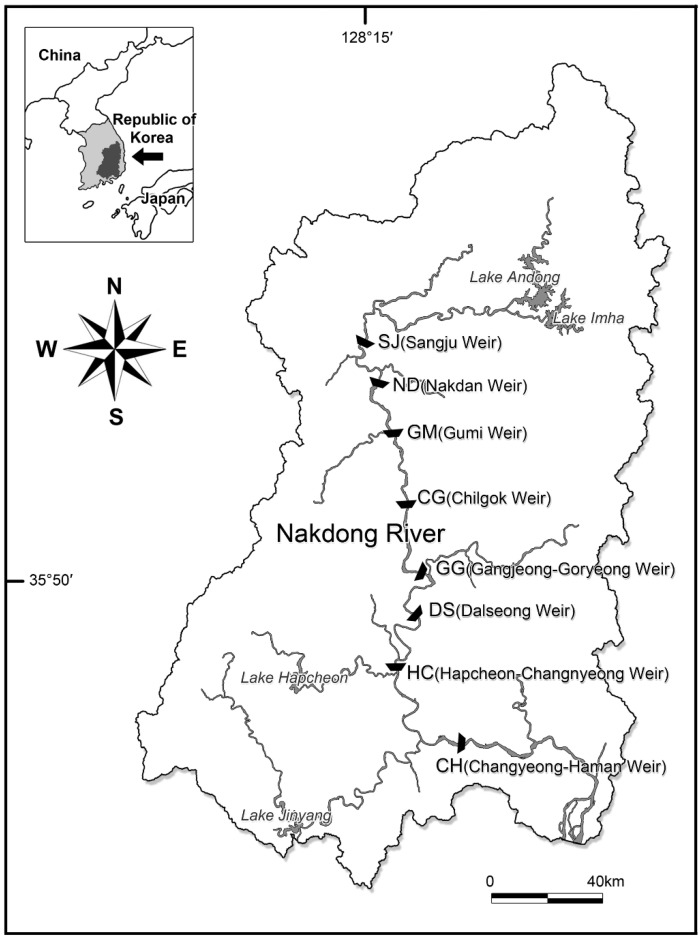
Sampling sites along Nakdong River, Korea. Black symbols indicate survey site for invasive nostocalean cyanobacteria.

**Table 1 toxins-14-00294-t001:** Intracellular contents of target gene (genus-specific) copy number in a cell of invasive nostocalean cyanobacteria.

No.	*Cuspidothrix* (rpoB)		*Sphaerospermopsis* (rbcLX)		*Cylindrospermopsis* (rpoC1)	
	Strain No.	Copies Cell^−1^	Strain No.	Copies Cell^−1^	Strain No.	Copies Cell^−1^
1	NIVA-711	1.4	NRERC-600	1.6	CS-1101	2.3
2	NRERC-650	1.0	NRERC-601	2.8	NRERC-501	1.2
3	NRERC-651	1.0	NRERC-602	4.1	NRERC-502	1.3
4	NRERC-652	1.1	NRERC-603	1.6	NRERC-503	1.3
5	NRERC-654	1.1	NRERC-604	3.3	NRERC-504	1.5
6	NRERC-655	1.1	NRERC-605	1.7		
7	NRERC-656	1.4	NRERC-606	4.1		
8	NRERC-657	1.3	NRERC-607	4.3		
9	NRERC-658	1.0	NRERC-608	3.7		
10	NRERC-659	1.1				
11	NRERC-660	1.2				
12	NRERC-661	1.0				
	Average	1.1	Average	3.0	Average	1.3

**Table 2 toxins-14-00294-t002:** Intracellular contents of target gene copy number and cyanotoxin concentration for three cyanobacterial strains.

Strain No.	Species Name	Target Gene	Cell Quota	
			Gene (Copies Cell^−1^)	Toxin (pg Cell^−1^)
NIVA-711	*Cuspidothrix issatschenkoi*	*anaF*	2.1	0.012
CS-1101	*Cylindrospermopsis raciborskii*	*cyrA*, *J*	2.0	0.013
NIVA-851	*Aphanizomenon gracile*	*sxtA*, *I*	2.1	0.001

**Table 3 toxins-14-00294-t003:** Result of toxin analysis using the enzyme-linked immunosorbent assay (ELISA) kit for samples collected at eight weir sites in Nakdong River (µg L^−1^).

Site	Saxitoxin ^1^			Anatoxin-a ^1^		Cylindrospermopsin ^1^
	13 Apr.	27 Apr.	11 May	1 Jun.	8 Jun.	Mar. to Nov.
SJ	ND ^2^	ND	ND	ND	ND	ND
ND	ND	ND	ND	0.174	0.283	ND
GM	ND	ND	ND	ND	0.154	ND
CG	ND	0.032	ND	ND	ND	ND
GG	ND	ND	0.024	ND	ND	ND
DS	0.043	ND	ND	ND	ND	ND
HC	ND	ND	ND	ND	ND	ND
CH	0.023	ND	ND	ND	ND	ND

^1^ Detection limit: saxitoxin 0.020 µg L^−1^, anatoxin-a 0.150 µg L^−1^, cylindrospermopsin 0.05 µg L^−1^. ^2^ ND: not detected.

**Table 4 toxins-14-00294-t004:** List of invasive nostocalean cyanobacteria and toxin-producing cyanobacteria cultures.

Species Name	Strains No.	Toxin
*Cylindrospermopsis raciborskii*	CS-1101	Cylindrospermopsin
*Cylindrospermopsis raciborskii*	NRERC-501, 502, 503, 504	Nontoxic
*Sphaerospermopsis* *aphanizomenoides*	NRERC-600, 601, 602, 603, 605, 606, 607	Nontoxic
*Sphaerospermopsis reniformis*	NRERC-604, 608	Nontoxic
*Cuspidothrix issatschenkoi*	NRERC-650, 651, 652, 655, 656, 657, 658, 659, 660	Nontoxic
*Cuspidothrix issatschenkoi*	NIVA-711, NRERC-654, 661	Anatoxin-a
*Aphanizomenon gracile*	NIVA-851	Saxitoxin

**Table 5 toxins-14-00294-t005:** List of ddPCR primers and probes for amplification of *rpoC1* (*Cylindrospermopsis*), *rbcLX* (*Sphaerospermopsis*), and *rpoB* (*Cuspidothrix*) genes.

Gene	Primer	Sequence (5′→3′)	References
*rpoC1*	dd-*rpoC1*-probe	ATCCTGGTAATGCTGACACACTCGTTT	This study
	dd-*rpoC1*-F	TGAGCAAATCGTCTACTTTAACTC	
	dd-*rpoC1*-R	GCACCAATTCCTACTTCTACC	
*rbcLX*	dd-sph-probe	TGTTTTGGCGCAGCTAGGCGA	This study
	dd-sph-F	ATCTATGGGGCTGGGTCAAG	
	dd-sph-R	GAATTTTCCCGGCAGAAAAG	
*rpoB*	dd-cus-probe	AACTGACAACGAGCAACAAACAACTGA	This study
	dd-cus-F	TAGTCAGTGGTCAATAGTCA	
	dd-cus-R	TCTCACCAATGGTTTTTGATT	

**Table 6 toxins-14-00294-t006:** List of ddPCR primers and probes for amplification of cyanotoxin genes (ana: anatoxin-a, cyr: cylindrospermpsin, sxt: saxitoxin).

Toxin	Gene	Primer	Sequence (5′→3′)	References
Ana	*anaF*	dd-*anaF*-probe	AAGCGCGGATGCTTGCAACC	This study
		dd-*anaF*-F	AAAGAATCCGACCTAGCTTT	
		dd-*anaF*-R	AACCTTCACTGCGAACATAC	
Cyr	*cyrA*	dd-*cyrA*-probe	AATTGCCAACCGTTATCCATGAAGAGTT	This study
		dd-*cyrA*-F	CCCATCCCACATTGATTGTAC	
		dd-*cyrA*-R	GCAGAACATAGGCATCTCATC	
	*cyrJ*	dd-*cyrJ*-probe	CTGATTCGCCAACCCAAAGAAATGCTCT	
		dd-*cyrJ*-F	GCATCAAGCGTATCATTTAAT	
		dd-*cyrJ*-R	AGCCTGTTTCTTCAAAGGTAAA	
Sxt	*sxtA*	dd-*sxtA*-probe	CTCCTCCCGACACATGGAACCC	This study
		dd-*sxtA*-F	GCTACACAACGAGCAACG	
		dd-*sxtA*-R	GGACGGTAATTAGCAATAATTCCC	
	*sxtI*	dd-*sxtI*-probe	TGAATATGGACTACTTCAACTACACGGT	
		dd-*sxtI*-F	CCATCTGTTGGATCTCAAAGA	
		dd-*sxtI*-R	TGTGGAACTTATGATTGGTCA	

## Data Availability

Not applicable.

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
