# Peer review of "Assessment of the Appearance and Toxin Production Potential of Invasive Nostocalean Cyanobacteria Using Quantitative Gene Analysis in Nakdong River, Korea"

_toxins, 2022, doi:10.3390/toxins14050294_

Round 1
Reviewer 1 Report
The manuscript entitled “Assessment of the appearance and toxin production potential of invasive nostocalean cyanobacteria using quantitative gene analysis in the Nakdong River, Korea” aimed to calculate the gene copy number per cell for each target gene through quantitative gene analysis based on genus-specific primers of the genera Cylindrospermopsis, Sphaerospermopsis, and Cuspidothrix and toxin primers of anatoxin-a, saxitoxin, and cylindrospermopsin. Authors concluded that quantitative gene analysis using toxin- and genus-specific primers enabled the precise analysis of the appearance times, cell densities, and toxin production potential of target algae.
Authors could discuss in brief the significance of their results in the context of public health, especially in light of today’s climate change, which favours harmful algal blooms and the growth of toxin producing algae with a presumed negative impact on the public health of coastal populations.
Suggested literature:
Fu, F., Tatters, A., Hutchins, D., 2012. Global change and the future of harmful algal blooms in the ocean. Mar. Ecol. Prog. Ser. 470, 207–233.
Gobler, C.J., 2020. Climate Change and Harmful Algal Blooms: Insights and perspective. Harmful Algae 91, 101731.
Griffith, A.W., Gobler, C.J., 2020. Harmful algal blooms: A climate change co-stressor in marine and freshwater ecosystems. Harmful Algae 91, 101590.
Moore, S.K., Trainer, V.L., Mantua, N.J., Parker, M.S., Laws, E.A., Backer, L.C., Fleming, L.E., 2008. Impacts of climate variability and future climate change on harmful algal blooms and human health. Environ. Heal. 7, S4.
Townhill, B.L., Tinker, J., Jones, M., Pitois, S., Creach, V., Simpson, S.D., Dye, S., Bear, E., Pinnegar, J.K., 2018. Harmful algal blooms and climate change: exploring future distribution changes. ICES J. Mar. Sci. 75, 1882–1893.
Trainer, V.L., Moore, S.K., Hallegraeff, G., Kudela, R.M., Clement, A., Mardones, J.I., Cochlan, W.P., 2020. Pelagic harmful algal blooms and climate change: Lessons from nature’s experiments with extremes. Harmful Algae 91, 101591.
Wells, M.L., Trainer, V.L., Smayda, T.J., Karlson, B.S.O., Trick, C.G., Kudela, R.M., Ishikawa, A., Bernard, S., Wulff, A., Anderson, D.M., Cochlan, W.P., 2015. Harmful algal blooms and climate change: Learning from the past and present to forecast the future. Harmful Algae 49, 68–93.
WHOI, 2022. Harmful Algal Blooms Understanding the threat and the actions being taken to address it. Woods Hole Oceanographic Institution, Woods Hole, Massachusetts.
Minor remarks:
Page 10, line 233 – change to “In this study we aimed to develop a quantitative gene analysis method…”
Page 10, line 244 – change to “…did not consider to detect target cyanobacteria in low-density samples”
Reviewer 2 Report
I revise this paper, in which quantitative gene analysis was used to evaluate toxin production potential of invasive cyanobacteria in real case study. In my opinion the structure of the paper is good, tables and figures are clear. However, several points should be addressed before possible publication on this Journal. I suggest major revisions and my comments are the following:
- In the introduction, I suggest to insert a brief description of cyanobacteria effects on human health to better highlight the importance of this study. Cyanotoxins could be a serious problem if present in drinking water. I suggest you a recent paper on this topic, https://doi.org/10.3390/toxins12120810
- In the introduction, try also to highlight what current gaps in literature will be overcome thanks to this work. What is the novelty? Please, better explain.
- Figure 1 and Figure 2: Are red points the means of repeated tests or not? If yes, confidence bars and number of repetitions should be provided.
- I think that the importance of your findings should be better discussed in the section 3 and resumed in section 4. Try to answer questions as: What can be possible practical applications of your findings? What can be the advantages? Who are the main stakeholders of your results? What can be further developments of you research?
Reviewer 3 Report
- Describe the temperate region. What are the climate conditions? is there any relation between both the temperature and precipitation with Toxin-producing cyanobacteria?
- in the abstract, check the mL-1 units.
- anaF (line 20 and 43) should be italic. Similarly, the genera names should be italic throughout the text.
- Line 55-56, ... difficult to analyze using a microscope - It's repeated after line 47-48.
- 11 of the 49 citations were currently published papers, however, the rest of them were not recently published. Use more recently published papers.
- You can add an extra section about the effects of toxin-producing cyanobacteria against microbial communities in aquatic and also on human&plants. So, why is it important, and how we can decrease their numbers?
Reviewer 4 Report
The manuscript submitted to Toxins and entitled “Assessment of the appearance and toxin production potential of invasive nostocalean cyanobacteria using quantitative gene analysis in the Nakdong River, Korea” provides information on properties of cyanobacteria species as essential for causing toxin production. The author showed the use of the gene copy number per cell for each target gene through quantitative analysis based on genus specific primers of invasive nostocalean cyanobacteria.
The manuscript has been prepared very carefully. Introduction and discussion in next chapters show the deep understanding of the issue. The descriptions of gene copy number and data analysis are described in details. The manuscript is written in a good scientific language and illustrated in an appropriate way. References used in the article were correctly selected to presented problems but I think that the authors should supplement bibliographic revision. There are new researches on Cylindrospermopsis raciborski ecology which can be useful for Authors’ data discussion, for example publications of Kokocinski M.
Reviewer 5 Report
This paper presents the cell density and spatial-temporal distribution of three invasive nostocalean cyanobacteria at eight sites along the Nakdong River, Korea, in 2020. Two procedures are compared. The first is the quantitative gene analysis and the second is the microscopic analysis. The influence of the water temperature is also presented. The analysis is quite complete and merits to be published in Toxins after the introduction of some minor changes.
Are the weirs the origin of these bacteria?
Figure 3 compares the quantitative gene and microscopic analyses. Although the agreement is noticeable, some differences may be observed. The authors could explain possible reasons of such disagreement and quantify it.
Future research lines in this field could be introduced.
Round 2
Reviewer 2 Report
I revise the paper and I found that authors amended the manuscript according to my suggestions. Therefore, I suggest the publication of this work.